# Obstetric Outcomes in the Surviving Fetus after Intrauterine Fetal Death in Bichorionic Twin Gestations

**DOI:** 10.3390/children8100927

**Published:** 2021-10-16

**Authors:** María de la Calle, Jose L. Bartha, Henar Serrano, David Ramiro-Cortijo

**Affiliations:** 1Obstetrics and Gynecology Service, Hospital Universitario La Paz, Paseo de la Castellana 261, 28046 Madrid, Spain; maria.delacalle@uam.es (M.d.l.C.); joseluisbartha@me.com (J.L.B.); 2Anesthesiology and Resuscitation Service, Hospital General Universitario de Toledo, Avenida del Río Guadiana, 45007 Toledo, Spain; henarserranomartin@hotmail.com; 3Department of Physiology, Faculty of Medicine, Universidad Autónoma de Madrid, C/Arzobispo Morcillo 2, 28049 Madrid, Spain

**Keywords:** twin pregnancy, bichorionic biamniotic, intrauterine fetal death, maternal complications, fetal complications

## Abstract

Twin pregnancies are high-risk gestations that increase the odds of obstetrical complications. They can also present specific and rare complications such as single intrauterine fetal death (IUFD). This complication has been extensively studied in monochorionic but not in bichorionic gestations. Today, the repercussions of IUFD may have on the surviving fetus, mother and bichorionic pregnancy are not known. Our objective was to study materno-obstetrical, fetal, and immediate delivery neonatal complications in bichorionic twin gestations with single IUFD compared to those with both fetuses alive. A retrospective and observational case-control study was performed in bichorionic biamniotic twin pregnancies, 22 complicated with single IUFD after 14 weeks (cases; IUFD group) and 51 with both fetuses alive (controls; non-IUFD group, from Obstetrics Service of La Paz Hospital (Madrid, Spain). The data were collected from obstetrical records. No significant differences were found in the rates of gestational diabetes, gestational hypertension, preeclampsia, neonatal complications, and prematurity between IUFD and non-IUFD groups. Statistical differences were found for the incidence of intrauterine growth restriction in the surviving fetus compared to first fetus of pregnancy with both fetuses alive (22.7% versus 2.0%, respectively; *p*-value = 0.012). There were no differences compared to second fetus (11.8%; *p*-value = 0.23). There was a high C-section rate in both groups (IUFD = 63.6%, non-IUFD = 64.7%; *p*-value = 0.19). In conclusion, single IUFD in bichorionic biamniotic twin gestations is a rare complication that should be closely monitored. It is essential that these gestations be attended by a clinical multidisciplinary team.

## 1. Introduction

In recent decades, Spanish society has changed socioeconomically, delaying motherhood. The Spanish Institute of Statistics found that the average maternal age of first pregnancy was 32 years old in 2018 [1]. Advanced maternal age for pregnancy (over 35 years) constitutes an independent risk factor increasing the odds of multiple gestation. In addition, multiple pregnancies can be increased by the use of assisted reproduction techniques (ART) [2], which are also more common with increasing maternal age [3]. According to the Spanish Fertility Society and other authors, two-thirds of multiple pregnancies are bichorionic biamniotic twin gestations that are achieved by ART [4,5].

Overall, twin pregnancies are complex gestations that must be managed in a high-risk obstetrical unit. Generally, the likelihood of obstetrical complications increases in twin pregnancies. In addition, they present specific complications such as intrauterine fetal death (IUFD) of one of the fetuses [6,7]. Usually, if IUFD occurs before 14 weeks of gestation, the embryo disappears, which is called evanescent fetal syndrome. If it occurs between 14 to 20 weeks, the fetus does not disappear; it reduces in size and volume, called “papyraceous fetus”. If fetal death occurs after 20 weeks, the fetus remains inside the uterus until the end of gestation [6,8]. This complication could be considered rare, however, given the increase in ART and consequently the increase of twins, it is more and more frequent. Currently, the prevalence of IUFD of one fetus is 0.5–6.8% of twin gestations during the second and third trimester [6,7,9].

After the death of one of the fetuses in utero, the factors that most decisively influence the perinatal prognosis of the surviving fetus are chorionicity and gestational age [7,10]. In monochorionic gestations, IUFD of one of the fetuses is more frequent than in bichorionic gestations due to the presence of a single placenta shared by both fetuses. Complications such as feto-fetal transfusion syndrome, anemia-polycythemia sequence and fetal growth restriction are responsible for the majority of IUFDs in monochorionic twin gestations [7]. The complications associated with IUFD in monochorionic gestations have been extensively studied as they are higher risk than bichorionic. It has been shown that IUFD in monochorionic gestations leads to higher rates of prematurity, neurological neonatal damage, and even neonatal death of the surviving fetus [11].

There are two theories that explain these results. First is the thrombotic components from the circulation of the dead fetus to the surviving fetus through vascular anastomoses, which would lead to alterations in coagulation. Second, the more recent and accepted theory, is the hemodynamic imbalance in which there would be a fast transfusion of the fetus’s blood from the surviving fetus into the circulation of the dead fetus through the placental anastomoses, with periods of hypoperfusion, hypotension and acute fetal anemia in the surviving fetus, resulting in multi-organ damage, especially neurological [4,6,7,8,9,12].

Bichorionic biamniotic twin gestations do not share neither placenta nor amniotic sac and have a lower rate of complications and fetal death, although they are not free of them. Theories that explain the complications of the surviving fetus in monochorionic gestations would not explain the complications associated with IUFD in bichorionic gestations. Furthermore, in bichorionic gestations, after a spontaneous or selective fetus death, the surviving fetus must develop in a hostile environment with a decomposing dead co-fetus for several weeks of gestation, sharing the same uterus with the alive co-fetus. At present, we do not know the repercussions that this complication may have on the surviving fetus, mother, and pregnancy.

Therefore, the relevance of this work lies in that it is the first study, to our knowledge, in which only bichorionic biamniotic twin gestations with IUFD are compared to bichorionic biamniotic twin gestations with both live fetuses. Our hypothesis was that bichorionic biamniotic twin pregnancies complicated by IUFD of one fetus after 14 weeks have more obstetric, maternal, fetal, and neonatal complications than bichorionic biamniotic twin gestations with both fetuses alive. Our aims were to compare materno-obstetrical, fetal, and immediate neonatal complications in bichorionic biamniotic twin gestations with the surviving fetus with respect to bichorionic biamniotic twin gestations with both fetuses alive.

## 2. Materials and Methods

### 2.1. Selection of the Cohort and Study Design

A retrospective observational case-control study of bichorionic biamniotic twin gestations complicated by IUFD from 14 week of gestation, monitored in the Obstetrics Service of Hospital Universitario La Paz (HULP; Madrid, Spain) between 2014 and 2019 was performed. A total of 22 bichorionic biamniotic twin gestations with one intrauterine fetal death (IUFD) and 51 bichorionic biamniotic twin gestations with both fetuses alive (non-IUFD) were included. Pregnant women were matched by maternal age and method of conception (spontaneous versus ART), making both groups as homogeneous as possible related to potential factors of IUFD.

The inclusion criteria for all women were: bichorionic biamniotic twin gestation and pregnancy follow-up at HULP. In addition, women included as the IUFD group (cases) must meet the diagnoses of IUFD of one fetus at least from above 14 weeks of gestation. To be included as non-IUFD group (controls), both fetuses must be alive during the entire gestation. Cases with death before 14 weeks of gestation were excluded, as well as those pregnancies that had no obstetrical control visits, and monochorionic twin pregnancies.

The study was carried out in accordance with the Declaration of Helsinki (2008 revision) and was approved by the Research Ethics Committee of HULP (PI-3890).

### 2.2. Data Collection

The data were collected from obstetrical records, including:

**Maternal variables and obstetrical complications**: Maternal age (years old), method of conception (spontaneous/ART). Following the HULP protocols, we recorded gestational diabetes (defined by oral glucose overload test), gestational hypertension (blood pressure higher than 140/90 detected after 20 weeks of gestation), preeclampsia (gestational hypertension and proteinuria in urine), threat of preterm delivery (presence of uterine contractions and/or cervical change before 37 weeks of gestation), premature rupture of membranes (PROM; rupture of the amniotic sac before 37 weeks of gestation).

**Fetal variables**: The variables recorded for the dead fetus were gestational age when death happened, and motive of fetal death categorized as spontaneous or selective feticide (for severe anomaly and genetical disorder, injecting KCl by cordocentesis into the fetus’ heart until asystole). In addition, in the surviving fetus we measured fetal growth restriction (fetus below 3rd percentile or below 10th percentile with hemodynamic alterations on Doppler), and small for gestational age (fetus below percentile 10th without hemodynamic alterations on Doppler).

**Labor variables**: Gestational age (weeks), preterm delivery (delivery before 37 weeks of gestation) and route of delivery. In addition, the variables studied in the newborn were weight (grams), pH of the umbilical artery, Apgar at first and at five minutes of life, admission to the Neonatal Intensive Care Unit (NICU), and neonatal death.

### 2.3. Statistical Analysis

The quantitative variables were expressed as median and interquartile range [Q1, Q3] and qualitative variables as relative frequency (%). Mann–Whitney’s U test was used to independently compare sum rank differences between groups. The association between qualitative variables was tested by chi-squared test with Fisher’s correction if necessary. A *p*-value < 0.05 was considered statistically significant. Statistical analysis was performed with R software (version 3.6.0, 2018, R Core Team, Vienna, Austria) within R Studio interface using *rio*, *tidyverse*, and *dplyr* packages.

## 3. Results

The maternal age in the non-IUFD group was 36.0 [32.5, 39.0] (Max. = 48.0; Min. = 30.0) years old and in IUFD was 36.0 [35.0, 39.0] (Max. = 45.0; Min. = 32.0) years old, without statistical differences (*p*-value = 0.73). ART represented 49.3% (36/73) in the non-IUFD group and 20.5% (15/73) in IUFD. ART and IUFD were not associated (χ^2^ = 0.04; *p*-value = 0.84). These data were not surprising since the case-control selection was based on maternal age and use of ART.

The IUFD group consisted of 59.1% (13/22) spontaneous fetal deaths and 40.9% (9/22) selective feticides. The motives of selective feticide are described in the Table 1. There was no statistical difference in maternal age compared to the motive of IUFD (Spontaneous = 36.0 [36.0, 36.0] years old; Selective = 38.0 [38.0, 38.0] years old; *p*-value = 0.50). ART represented 45.5% (10/22) of spontaneous fetal death and 22.7% (5/22) of selective feticides. ART and motive of fetal death were not associated (χ^2^ = 0.35; *p*-value = 0.55).

The average gestational age of fetal death was 22.1 [18.3, 24.9] (Max. = 36.3; Min. = 14.0) weeks of gestation, being 66.6% below the limit of viability (24 weeks of gestation). The fetal death was spontaneous at an average of 24.0 [19.0, 31.0] (Max. = 36.3; Min. = 16.0) weeks of gestation and was selective at 21.4 [16.6, 23.0] (Max. = 24.7; Min. = 14.0) weeks of gestation, without significant differences (*p*-value = 0.10).

### 3.1. Materno-Obstetrical Complications

No statistical association was detected in any of maternal and obstetric complications between the non-IUFD and IUFD groups, nor between the motive for fetal death in the IUFD group (Table 2).

### 3.2. Fetal Complications

There was higher prevalence of fetal growth restriction in the surviving fetus of IUFD compared with the first fetus of non-IUFD. However, no statistical association was detected for second fetus or any fetus for small gestation age complications between non-IUFD and IUFD groups (Table 3). In addition, we did not detect association in fetal growth restriction nor small gestational age by the motive of fetal death (χ^2^ = 1.17, *p*-Value = 0.28; χ^2^ = 0.08, *p*-value = 0.77; respectively).

### 3.3. Labor and Gestational Age at Delivery

The average gestational age at delivery in the non-IUFD group was 36.7 [35.7, 37.4] (Max. = 38.7; Min. = 29.9) weeks of gestation and in the IUFD group was 36.4 [34.5, 38.0] (Max. = 41.0; Min. = 26.3) weeks of gestation, without statistical differences (*p*-value = 0.84). Prematurity was more prevalent in the non-IUFD group (45.3% (33/73)) than the IUFD group (16.4% (12/73)). However, prematurity and IUFD was not associated (χ^2^ = 0.31; *p*-value = 0.58). The route of delivery was 23.5% (12/51) vaginal, 11.8% (6/51) instrumental and 64.7% (33/51) C-section in non-IUFD, and 36.4% (8/22) vaginal and 63.6% (14/22) C-section in IUFD, without statistical association (*p*-value = 0.19). There was no instrumental delivery in the IUFD group.

There was no statistical difference in gestational age at delivery compared to the motive of IUFD (Spontaneous = 36.1 [35.3, 37.3] weeks; Selective = 37.3 [34.3; 38.0] weeks; *p*-value = 0.82). Prematurity was not associated with the motive of fetal death (Spontaneous = 36.3%, Selective = 18.2%, χ^2^ = 0.83, *p*-value = 0.43). C-section in surviving fetus was performed in 36.4% (8/22) of spontaneous fetal deaths and 27.3% (6/22) of selective feticides, without statistical differences (χ^2^ = 0.06, *p*-value = 0.81).

### 3.4. Immediate Neonatal Variables at Delivery

There was no antepartum fetal death in any cases. No statistical association was detected in any of the immediate neonatal variables at delivery between first or second fetus in non-IUFD group and surviving fetus in IUFD group (Table 4).

## 4. Discussion

In single pregnancies, late miscarriage and stillbirth produce uterus evacuation and pregnancy termination. These events in twin gestation with one fetal death do not happen, because the surviving co-fetus must continue gestating. To the best of our knowledge, this is the first study in which only bichorionic biamniotic twin gestations with a dead fetus were compared to bichorionic biamniotic twin gestations with both live fetuses. We could suggest that fetal death in bichorionic biamniotic twin pregnancies, even being an obstetrical complication, is not as serious as in monochorionic gestations because it did not increase maternal-obstetrical complications, prematurity, fetal complications, or antepartum fetal death of the surviving fetus. It seems that not sharing a placenta with the dead fetus could be a protective factor for the surviving fetus. This could reinforce the theory of hemodynamic imbalance as a pathophysiological mechanism of fetal damage in monochorionic gestations [4,12].

It is important to note that IUFD has a multifactorial etiology, not only with fetal genetic disorders and congenital malformations but also maternal infections and placental/umbilical cord complications [13,14,15]. The literature related to preventing it is usually controlling factors prior to conception, such as avoiding unhealthy habits (alcohol, tobacco, and illegal drugs consumption) and maintaining genital hygiene to prevent infections, and during pregnancy, as well as regular obstetrical controls to screen abnormal fetal malformation and pregnancy complications (i.e., gestational diabetes, gestational hypertension and preeclampsia), controlling pregnancy in advanced maternal age (maternal age ≥ 35 years), and increasing appropriate healthy habits with a balanced diet and physical activity [16,17,18], particularly in twin pregnancies.

In a study of monochorionic and bichorionic twin gestations between 2001 and 2006, maternal age was 30 years and 40% used ART in bichorionic gestations with a dead fetus [19]. Our data collected between 2014 and 2019 showed an increase in maternal age and the use of ART (overall, 36 years old and close to 70%, respectively), which led us to believe that there is a trend to increase maternal age and use of ART to achieve pregnancy. In the Fichera et al. study, ten cases of bichorionic gestation with spontaneous fetal death were described. Fetal death included fetal growth restriction, chromosomal alterations, placental pathology, and intrauterine infection [19]. In our study, most fetal deaths occurred spontaneously, however close to 41% were legal terminations of pregnancy due to chromosomal abnormalities and fetal malformations. This difference is because legal termination of pregnancy was not contemplated years ago in many countries and therefore is not described in the study of Fichera et al.

Regarding obstetric complications, it has been established that fetal death in monochorionic gestations leads to high rates of preterm delivery, neurological damage, and even neonatal death in the surviving co-twin [11]. In bichorionic gestations, due to the placenta not being shared, complications are less in the surviving co-twin. Of these complications, the threat of preterm delivery could be highly increased in bichorionic gestations [10,20]. This fact could be explained by uterine irritability triggered by the fetal death releasing pro-inflammatory molecules as it decomposes. However, this hypothesis needs to be deeply explored. In our work, the threat of preterm delivery was higher, but not significant, in the IUFD group (non-IUFD = 21.6% versus IUFD = 40.9%). Probably, if we would increase the sample size, we could detect the association.

Although we observed a greater tendency of threat of preterm delivery, there was no higher rate of PROM or prematurity, probably due to tocolytic administration following HULP obstetrical protocols. Uterine distention is directly related to the number of fetuses. Over distension of the uterus with two fetuses could increase contractions, PROM and, finally, premature delivery [21]. Although without significant differences, we detected higher prevalence of prematurity in twins than IUFD gestations. In addition, a twin gestation with a fetal death could be a pro-inflammatory environmental factor able to lead to contractions and premature rupture of membranes, compared to twin gestations with both fetuses alive; however, we did not find any differences. We observed in both groups that the gestational age at delivery was below 37 weeks. It was estimated that 60% of deliveries in bichorionic twin gestations occur prematurely [21,22].

Regarding maternal complications, they were similar in both groups probably due to the age-matched nature of the study. Advanced maternal age is a risk factor for gestational diabetes, gestational hypertension, and preeclampsia [3]. In fact, twin gestation is also a risk factor for these complications [2]. However, we did not observe a significant different rate of these complications between groups. Increasing the sample size could be necessary to detect differences.

The data of our study reflect a higher incidence of fetal growth restriction in the surviving twin (22.7%), especially compared with the literature where the prevalence of fetal growth restriction in developed countries is 3.5% [23]. In addition, it is more frequent in monochorionic than bichorionic twin gestations [23]. This complication can be explained by the hostile uterus environment in which the surviving fetus is developing, since the decomposition of the dead co-twin could increase of pro-inflammatory molecules delaying growth. These findings should help us to follow-up the surviving fetus closely and ultrasonographically in bichorionic biamniotic twin gestations and not to consider them as a single gestation. Furthermore, it is interesting that more fetal growth restriction is observed in the surviving fetus with respect to the first twin and not with respect to the second twin with both twins alive. Probably, the restriction of movements and a smaller placenta in the second twin could trigger its intrauterine growth restriction. Moreover, this fact is favored for advanced maternal age and the use of ART [23], characteristics described in our cohort. Prematurity in twin gestations related to fetal growth restriction was estimated to be 33% [24]. Although our data of prematurity and IUFD was not associated, overall, we observed close to 62% prematurity and 16% fetal growth restriction.

Finally, no cases of antepartum fetal death were observed. Fortunately, it did not seem that IUFD increased odds of this complication, probably related to the close clinical follow-up in a high-risk pregnancy unit. The only fetal complication that was increased was intrauterine growth restriction of the surviving fetus; thus, this surviving fetus should be clinically and closely monitored.

Overall, the proportion of C-sections was close to 64%; the literature shows that C-section does not improve perinatal prognosis of the deceased fetus in a single pregnancy, therefore IUFD is not considered an indication for C-section. Thus, vaginal delivery is allowed unless there are obstetric contraindications [8,10,20,25]. However, HULP presents high rate of C-sections in these gestations due to maternal desire, reinforcing that psychological support in the obstetrical care team is very necessary.

### Study Limitations and Future Directions

It is important to note that in this report we discuss the hypothesis of a pro-inflammatory uterine environment produced by a dead fetus that would condition both the pregnancy and the development of the surviving fetus in bichorionic twin pregnancies. However, we did not evaluate any biochemical parameter. Therefore, more studies would be needed to be able to confirm this assumption as well as enrolling a large cohort of cases.

In addition, the prevalence detected in this study could be treated as starting point to establish new follow-up protocols increasing the sample size, which does not consider IUFD as a single pregnancy. This reinforces the idea of close neonatal follow-up and continued maternal psychological support.

IUFD is a profoundly delicate situation for the parents who suffer it. When caring for a couple experiencing a stillbirth, it is important to convey empathy and sensitivity while being mindful of their emotional needs [16]. In this aspect, clinical psychologist teams need to be prepared to cover all emotional needs. In addition, a psychological follow-up appointment should be made as soon as possible for the parents. Postpartum depression and post-traumatic stress disorder are the most frequent conditions following IUFD [16]. Healthcare teams should answer immediate questions that a parent could have. Furthermore, the clinical psychologist needs to reinforce that the mother did nothing to cause this outcome, and, finally, all assistance teams should provide privacy and time to deal with emotions [16]. It is important to note that parents, particularly mothers, can experience the five stages of grief (denial, anger, bargaining, depression, and, finally, acceptance) back and forth [26,27]. In the diagnoses of IUFD with a survival co-twin, it could be helpful if the mother has a support person with her. Finally, parents with IUFD emotionally benefit from their healthcare providers using coping and empowerment skills, inquiring about their emotional needs, and providing information regarding mental health referrals [28,29]. Additionally, after the delivery, it must always be kept in mind that parents fight with two opposite feelings: sadness over the loss of a dead fetus and joy over the birth of the living co-fetus. That is why it is important to prepare parents psychologically from the diagnosis of IUFD, to better cope with this situation throughout pregnancy, during childbirth, and postpartum.

## 5. Conclusions

In conclusion, intrauterine fetal death in bichorionic biamniotic twin gestations is a rare, but increasing, complication due to the rise of these gestations, and should be closely monitored. According to our cases of bichorionic biamniotic twin gestations with one dead fetus compared to those with both live fetuses, there were no more maternal-obstetric complications nor neonatal complications immediately after delivery in the surviving fetus of IUFD gestations. However, there was a higher prevalence of fetal growth restriction in the surviving fetus. There was a higher rate of spontaneous compared to induced intrauterine fetal death. Most fetal deaths occurred before 24 weeks and most deliveries occurred before 37 weeks of gestation. The plan of action to control IUFD in twin gestations must always be individualized.

This work allows us to advise and inform parents, with research evidence, who are expecting a surviving neonate after the death of co-fetus, which is a stressful event that could disrupt the emotional balance of the family. Thus, it is essential that pregnancy be attended by a multidisciplinary team in a tertiary care center to provide obstetric, neonatal, and nursing assistance, and psychological support.

## Figures and Tables

**Table 1 children-08-00927-t001:** Motive and week of gestation of selective feticides.

Case	Maternal Age (Years)	Gestation Age (Weeks)	ART	Motive
1	38	23.4	Yes	Bilateral cavitation in ganglionic eminences and early fetal growth restriction
2	39	22.5	Yes	Rhizomelia of right arm, right equinovarus foot and pyelic ectasia
3	39	23.0	Yes	21 trisomy with mild ventriculomegaly
4	40	15.0	No	21 trisomy
5	36	23.0	No	Hemivertebrae, thoraco-columnar scoliosis, and left renal agenesis
6	40	21.4	Yes	Double opening right ventricle, vessels in transposition, rhizomelia of left arm
7	32	16.6	No	Megacystis
8	35	14	No	Megacystis
9	35	18	Yes	Left diaphragmatic hernia, occupation of left thorax by left hepatic lobe and intestinal loops

Assisted Reproduction Techniques (ART). Gestational age when fetal death happened.

**Table 2 children-08-00927-t002:** Maternal and obstetrical complications between groups.

	IUFD		*p*-Value	Motive of IUFD		*p*-Value
	No (*n* = 51)	Yes (*n* = 22)		Spontaneous (*n* = 13)	Selective (*n* = 9)	
Gestational DM	5.9% (3)	9.0% (2)	0.62	7.7% (1)	11.1% (1)	0.78
Gestational HT	2.0% (1)	0.0% (0)	0.51	0.0% (0)	0.0% (0)	-
Preeclampsia	7.8% (4)	4.5% (1)	0.61	7.7.% (1)	0.0% (0)	0.39
Threat of PT delivery	21.6% (11)	40.9% (9)	0.16	46.2% (6)	33.3% (3)	0.55
PROM	27.5% (14)	9.0% (2)	0.15	7.7% (1)	11.1% (1)	0.78

Data shown relative frequency and simple size (*n*). Intrauterine fetal death (IUFD); Diabetes Mellitus (DM); Hypertension (HT); preterm (PT); Premature Rupture of Membranes (PROM). *p*-Value was extracted by Chi-squared test.

**Table 3 children-08-00927-t003:** Fetal complications between non-IUFD fetus and surviving fetus of IUFD.

	Non-IUFD (*n* = 51)		IUFD (*n* = 22)	*p*-Value ^I^	*p*-Value ^II^
	Fetus I	Fetus II	Surviving Fetus		
Fetal growth restriction	2.0% (1)	11.8% (6)	22.7% (5)	0.012	0.23
Small for gestational age	7.8% (4)	25.5% (13)	13.6% (3)	0.74	0.26

Data shown relative frequency and simple size (*n*). Intrauterine fetal death (IUFD). *p*-Value was extracted by Chi-squared test. *p*-Value ^I^ compare surviving fetus with fetus I; *p*-Value ^II^ compare surviving fetus with fetus II.

**Table 4 children-08-00927-t004:** Immediate neonatal variables at delivery between non-IUFD fetus and surviving fetus of IUFD.

	Non-IUFD (*n* = 51)		IUFD (*n* = 22)	*p*-Value ^I^	*p*-Value ^II^
	Fetus I	Fetus II	Surviving Fetus		
Birth weight (g)	2400 [2095, 2725]	2250 [2012, 2570]	2488 [1866, 3070]	0.65	0.23
Umbilical artery pH	7.32 [7.28, 7.35]	7.31 [7.25, 7.34]	7.29 [7.26, 7.33]	0.27	0.82
Apgar at 1 min	9.0 [8.0, 9.0]	9.0 [8.0, 9.0]	9.0 [8.0, 9.0]	0.18	0.46
Apgar at 5 min	10.0 [9.5, 10.0]	10.0 [9.0, 10.0]	10.0 [9.0, 10.0]	0.06	0.89
NICU admission	47.1% (24)	58.8% (30)	45.5% (10)	0.90	0.29

Data shown median and interquartile range [Q1, Q3] in quantitative variables and relative frequency and simple size (*n*) in qualitative variable. Intrauterine fetal death (IUFD), Neonatal Intensive Care Unit (NICU). *p*-Value was extracted by Chi-squared test in qualitative variables and Wilcoxon sum rank test in quantitative variables. *p*-Value ^I^ comparing surviving fetus with fetus I; *p*-Value ^II^ comparing surviving fetus with fetus II.

## Data Availability

The data presented in this study are available on request from the corresponding author. The availability of the data is restricted to investigators based in academic institutions.

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
