# Peer review of "Obstetric Outcomes in the Surviving Fetus after Intrauterine Fetal Death in Bichorionic Twin Gestations"

_children, 2021, doi:10.3390/children8100927_

Round 1

Reviewer 1 Report

The authors have described the survey based study on twin pregnancies and written well in the manuscript.

1. It would be better if the authors discuss how to prevent such kinds of fetal complications. What precautions can be taken? 2. Is there any association to delayed pregnancy for IUFD? All the reported maternal ages are more than 35. What percentage of early pregnancy show IUFD?

Author Response

The authors have described the survey-based study on twin pregnancies and written well in the manuscript.

Response: Thank you for your time reviewing our article. Please see below our point-by-point responses.

  1. It would be better if the authors discuss how to prevent such kinds of fetal complications. What precautions can be taken?

Response: This is a great point, unfortunately the IUFD has a multifactorial etiology which, in many cases, does it unpredictable. Most of the literature to prevent it are about usual control during gestation and regular obstetrical visit. All these items were included in the text (lines 214-222).

  1. Is there any association to delayed pregnancy for IUFD? All the reported maternal ages are more than 35. What percentage of early pregnancy show IUFD?

Response: Although we found no association between IUFD and advanced maternal age, advanced maternal age is considered a risk factor for pregnancy, particularly in twin gestations. Twin gestation and advanced maternal age could predispose to IUFD, strongly conditioning pregnancy. The most literature of early pregnancy is related to stillbirth, that it is estimated in adjusted OR=1.3 [1.1-1.6] in pregnancy with 16-17 years compared to 24 years (PMID: 24641534). The literature related to IUFD with a survival co-twin of twin pregnancy in early pregnancies is poorly explored, because it is a rare condition.

Reviewer 2 Report

Thank you for the opportunity to review the work. It is properly carried out methodological, although it covers a small study group. The discussion is conducted correctly. If possible, please refresh the literature, because only 1/4 of the items are younger than 5 years

Author Response

Thank you for the opportunity to review the work. It is properly carried out methodological, although it covers a small study group. The discussion is conducted correctly. If possible, please refresh the literature, because only 1/4 of the items are younger than 5 years.

Response: Thank you for your time reviewing our article. This theme is poorly explored, being the literature old. Both aspects do the article more attractive in OBS/GYN filed. Therefore, it could active new research in the area. However, we agree with the reviewer’s comment, and we have added new information (refs. 13-18 and 26-29).